# Ultrasound-Assisted Extraction of Semi-Defatted Unripe Genipap (*Genipa americana* L.): Selective Conditions for the Recovery of Natural Colorants

Grazielle Náthia-Neves [1,2,*], Ádina L. Santana [1,3], Juliane Viganó [1,4], Julian Martínez [1] and Maria Angela A. Meireles [1,*]

1   School of Food Engineering, University of Campinas (UNICAMP), R. Monteiro Lobato 80, Campinas 13083-862, SP, Brazil; adina.santana@gmail.com (Á.L.S.); julianevigano@gmail.com (J.V.); julian@unicamp.br (J.M.)
2   Department of Chemical Engineering and Environmental Technology, University of Valladolid, Prado de la Magdalena 5, 47011 Valladolid, Spain
3   Food Science Institute, Kansas State University, 1530 N Mid Campus Drive, Manhattan, KS 66506, USA
4   Department of Chemical Engineering, Institute of Environmental, Chemical and Pharmaceutical Sciences, Universidade Federal de São Paulo, R. São Nicolau 210, Diadema 09913-030, SP, Brazil
*   Correspondence: grazinathia@yahoo.com.br (G.N.-N.); maameireles@lasefi.com (M.A.A.M.); Tel.: +55-19-3521-0100 (G.N.-N. & M.A.A.M.); Fax: +55-19-3521-4027 (G.N.-N. & M.A.A.M.)

**Abstract:** Ultrasound-assisted extraction (UAE) of semi-defatted unripe genipap (SDG) using supercritical $CO_2$ was performed to enhance the recovery of natural colorant iridoids genipin and geniposide. There are currently few natural sources of iridoids, and their application as colorants is scarce. The UAE resulted in extracts with blue and green colors using water and ethanol, respectively. The highest global yield and genipin content was recovered with water, and the geniposide was significantly recovered with ethanol. With water at 450 W, the UAE raised the maximum global yield (25.50 g/100 g raw material). At 150 W and 7 min, the maximum content of genipin (121.7 mg/g extract) and geniposide (312 mg/g extract) was recovered. The total phenolic content (TPC) and antioxidant capacity with the oxygen reactive antioxidant capacity (ORAC) assay were also high in aqueous extracts. Ethanolic extracts showed high ferric-reducing ability antioxidant potential (FRAP) values. UAE showed an efficient and fast method to obtain different extracts' fractions from SDG, which have a wide spectrum of applications, especially as natural food colorants.

**Keywords:** ultrasound-assisted extraction; iridoids; genipin; geniposide; antioxidants

## 1. Introduction

Genipap (*Genipa americana* L.) is commonly used to produce jams, ice cream, soft drinks, liquor, and wine, consumed by the populations in the Brazilian Amazon. Unripe genipap contains genipin and geniposide (which are not present in the ripe fruits). These compounds have been recently associated with health benefits, such as antiviral [1] and anti-allergic [2] activities, as well as neuroprotection [3,4] and antidiabetics [5]. Genipin is responsible for the violet to the blue-dark complex coloring, and geniposide is attributed to the yellow-green complex [6].

Genipin can quickly react with amino acids in the presence of oxygen to produce a blue color complex [7]. Geniposide represents more than 70% of the iridoid content in the unripe genipap [8]. To obtain the blue-color complex, geniposide requires hydrolysis, often mediated by enzymes [8–10].

There is a need for natural colorants to replace the synthetic ones in food products such as beverages, desserts, gels, and confectionery. The color of food is often associated with the flavor, safety, and nutritional value of the product [11]. Therefore, these compounds' obtention from natural sources represents an important alternative to provide additives



to the food industry. Indeed, the extraction of iridoids from genipap is common with organic solvents such as ethyl acetate and n-butanol or enzymatic hydrolysis with β-glucosidase [12].

Moreover, the low stability of natural colorants and other antioxidants in environmental conditions induces the need for rapid extraction processes of these target compounds, as well as the use of Generally Recognized as Safe (GRAS) solvents (water and ethanol). Recent studies recovered genipin and geniposide from crude unripe genipap using pressurized liquid extraction with ethanol [13] and enzyme-assisted extraction combined with high-pressure processing [14].

The use of supercritical fluid extraction (SFE) as a type of pre-treatment (by the recovering of oil from a plant matrix) carried out on the raw material before the extraction process of polar compounds has already been reported in the literature. SFE, for instance, has already been combined with other techniques such as pressurized liquid extraction, pressurized hot water extraction, low-pressure solvent extraction, and ultrasound-assisted extraction to obtain polyphenols, iridoids, flavonoids, and polar pigments [15–17]. Ultrasound-assisted extraction (UAE) has been shown as an efficient technique to recover bioactive compounds from plant matrices because of induced acoustic cavitation that disrupts cell walls and enhances the diffusion of target compounds into the solvent [18]. Considering extractions from by-product fractions, UAE was effective in recovering lycopene [19] and pectin [20] from tomato and phenolic compounds from lime peels [21].

Ramos-de-la-Peña et al. [22] used UAE of crude genipap coupled with enzyme-assisted extraction to enhance the obtaining of genipin. Strieder et al. [18] used UAE of crude genipap with the use of milk as a solvent to provide a blue-colored carrier system.

In the current work, we used, for the first time, UAE as an alternative process to recover iridoids from semi-defatted unripe genipap fruits (SDG), a genipap by-product derived from lipid extraction using supercritical carbon dioxide (SC-CO₂). The results obtained from this work are expected to support further development of processes to intensify the extraction and stability of natural colorants from genipap until the complete valorization of this plant for food and non-food applications.

## 2. Material and Methods

### 2.1. Sample Preparation

Crude unripe genipap fruits (*Genipa americana* L.) were provided by the company Sítio do Bello (Paraibuna, São Paulo, Brazil), transported and stored under freezing conditions ($-18\,°C$) until drying in a freeze-dryer (LP1010, Liobrás, São Carlos, SP, Brazil) at 300 μm Hg for 96 h. The dried fruits (whole fruit with peel) were ground in a knife mill (Marconi, model MA-340, Piracicaba, Brazil), and the particle size distribution was determined in a vibratory system (Bertel, model 1868, Caieiras, Brazil) using sieves from 16 to 80 mesh (Tyler series, Wheeling, IL, USA). The mean particle diameter (dp) was $0.23 \pm 0.03$ mm, determined according to the method proposed by ASAE [23]. The ground genipap was semi-defatted (the nonpolar fraction of the raw material was reduced from 8 to 0.9 wt.%) by SFE using $CO_2$ as the solvent. The SFE process was performed according to the method described by Nathia-Neves et al. [15], which consists of continuous solvent flow through a fixed solid bed of sample particles placed inside the extraction column. The SFE conditions were at the temperature of 60 °C, the pressure of 300 bar, and the solvent to raw material ratio of 16 ($w/w$). The SDG was stored at $-18\,°C$ in the absence of light before further experiments.

The SDG was characterized according to moisture (method 920.151 from AOAC [24]): $13.9 \pm 0.1$ wt.% dry basis (d.b.), ash (method 923.03 from AOAC [24]): $4.4 \pm 0.3$ wt.% d.b., protein (method 970.22 from AOAC [24]): $10 \pm 1$ wt.% d.b., lipids (method 963.15 from AOAC [24]): $0.9 \pm 0.3$ wt.% d.b., and carbohydrates (calculated by difference): 84.7 wt.% d.b.

### 2.2. Reagents

Ethanol (99.0%, Dinâmica, Diadema, Brazil) and distilled water (Millipore, Bedford, USA) were used as solvents. The iridoids standards (genipin > 98% and geniposide > 98%)

were obtained from Sigma-Aldrich (Sao Paulo, Brazil). Acetonitrile (HPLC grade, J. T. Baker, Phillipsburg, NJ, USA), ultrapure water (Millipore, Bedford, MA, USA), and formic acid (Dinâmica, Diadema, Brazil) were used in the high-performance liquid chromatography (HPLC). The Folin–Ciocalteu reagent and gallic acid (Sigma-Aldrich, Sao Paulo, Brazil) were used for the total phenolic content analysis. The reagents used in the assays to measure antioxidant capacity were 6-hydroxy-2,5,7,8-tetramethylchromane-2-carboxylic acid (Trolox), ferric chloride ($FeCl_3$) 2,4,6-tris(2-pyridyl)-s-triazine (TPTZ), 2,2'-azobis(2-methylpropionamidine) dihydrochloride (APPH), and fluorescein, which were purchased from Sigma-Aldrich (Sao Paulo, Brazil).

### 2.3. Experimental

The UAE was performed with a 19 kHz ultrasonic probe (Unique, 800 W, Indaiatuba, Brazil). The effects of nominal power (150, 300, and 450 W), extraction time (1, 3, 5, and 7 min), and solvent (ethanol and water) were evaluated. Approximately 1.5 g of raw material was inserted in a 50 mL falcon tube containing 25 mL of solvent. The probe contact height with the medium was standardized to 25 mm. An ice bath was used to prevent the overheating of the extraction medium. The extracts were separated from the solid fraction using 0.22 mm Whatman paper filters and subsequently stored at −18 °C in the absence of light until further analyses.

### 2.4. Extract Evaluation

#### 2.4.1. Global Yield

The global yield ($X_0$) of SDG extracts was determined by removing the solvent from the extract by evaporating a 5 mL aliquot in an oven (TE-395, Tecnal, Piracicaba, Brazil) at 100 °C for 24 h. The global yield for each UAE condition was calculated according to Equation (1):

$$X_0 = \left( \frac{m_{EXT}}{F_0} \right) \times 100 \tag{1}$$

where $X_0$ (% (g extract/100 g SDG), dry basis) is the global extraction yield, $m_{EXT}$ (g) is the mass of dried extract, and $F_0$ (g) is the mass of raw material used in UAE.

#### 2.4.2. Total Phenolic Content (TPC)

The extracts' total phenolic content (TPC) was quantified using the Folin–Ciocalteu reagent [25]. Each extract was diluted in distilled water. Triplicates of the sample (20 μL) and Folin–Ciocalteu (20 μL) reagent were mixed and incubated at room temperature for 3 min. Afterwards, 20 μL of sodium carbonate solution and 140 μL of distilled water were added and the mixture was kept in the dark for 2 h at room temperature. The absorbance was read at 725 nm with the aid of a microplate reader (FLUOstar Omega, BMG LABTECH GmbH, Ortenberg, Germany). The gallic acid standard (0.02 to 0.16 mg/mL) was used, and the results were expressed as mg gallic acid equivalent (GAE) per g of SDG extract.

#### 2.4.3. Iridoids Quantification by HPLC Analysis

The iridoids (genipin and geniposide) were quantified by HPLC coupled with a diode-array detector (HPLC-DAD, Waters, Alliance model E2695, Milford, CT, USA) system and a C18 column (Kinetex, 100 × 4.6 mm i.d.; 2.6 μm; Phenomenex, Torrance, CA, USA) at 35 °C. The iridoids were detected according to the method described by Náthia-Neves et al. [26]. The mobile phases consisted of water (A) and acetonitrile (B), both acidified with 0.1% formic acid ($v/v$). The elution gradient at 1.5 mL/min was performed as follows: 0 min: 99% (A), 9 min: 75% (A), 10 min: 99% (A), and 13 min: 99% (A). The calibration curves of the iridoids were obtained in the range of 0.1–1000 μg/mL for geniposide ($R^2 = 0.9998$) and 0.1–625 μg/mL for genipin ($R^2 = 0.9998$). Figure 1 shows the chromatograms of the standards and the extracts obtained from SDG. Each sample was injected in duplicate, and the results were expressed in mg of genipin or mg of geniposide per g of dried SDG extract.

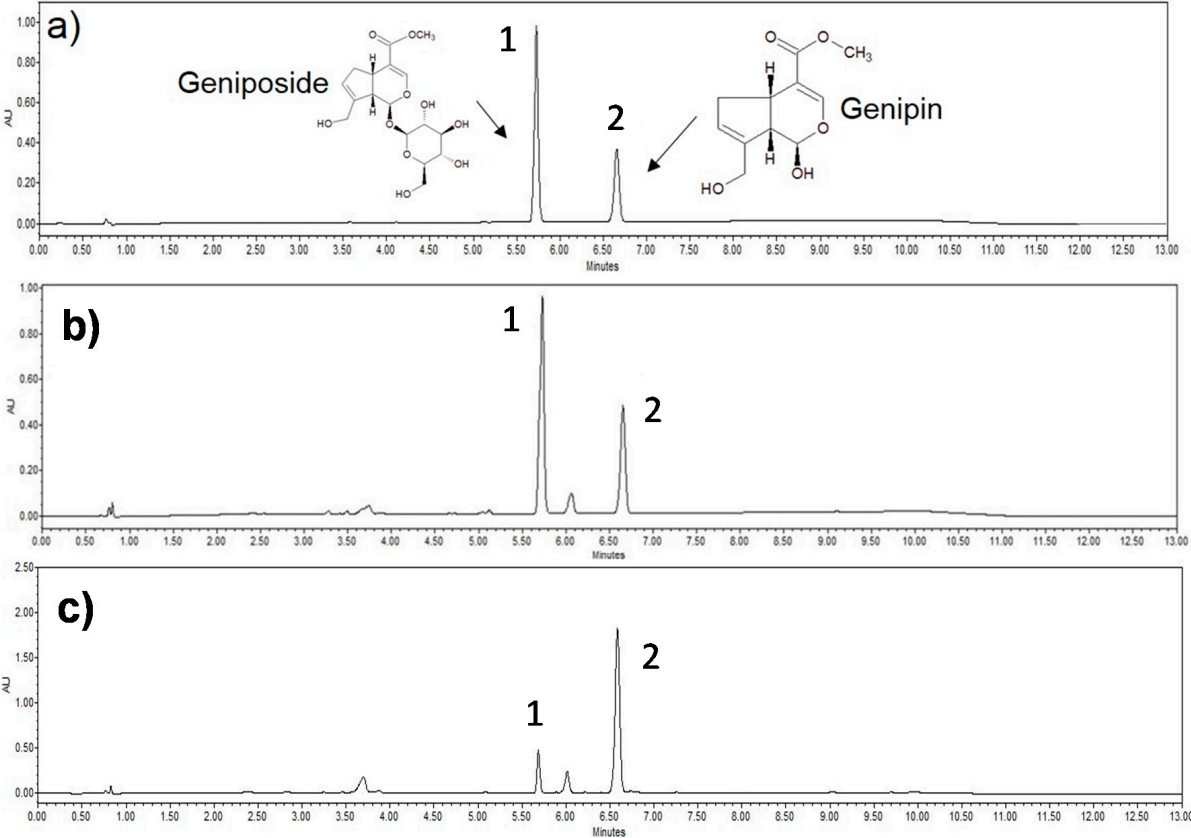

**Figure 1.** Representative HPLC/DAD chromatograms for genipin and geniposide recovered at 240 nm: (**a**) standard solution of genipin and geniposide, (**b**) ethanolic extract from SDG obtained at 150 W and 7 min, and (**c**) aqueous extract from SDG obtained at 150 W and 7 min. The retention times of geniposide and genipin were 5.73 and 6.65 min, respectively.

### 2.4.4. Antioxidant Capacity

The ferric-reducing ability of extracts was determined by the FRAP method [27]. The FRAP solution was prepared using 0.3 M acetate buffer (pH 3.6), 10 mM TPTZ in 40 mM HCl solution, and 20 mM $FeCl_3$ (10:1:1, $v/v/v$). Triplicates of the extract diluted in water (25 μL) were added to 175 μL of FRAP solution and kept in the dark for 30 min at 37 °C. The absorbance was measured at 595 nm using a microplate reader (a FLUOstar Omega BMG LABTECH GmbH, Ortenberg, Germany). The Trolox standard curve was obtained (0.01 to 0.06 mg/mL). Results were expressed in mg Trolox equivalent (TE) per g of SDG extract.

The oxygen radical absorbance capacity (ORAC) was measured by diluting the extracts and Trolox (5–25 μg/mL) in a 75 mM potassium phosphate buffer at pH 7.4 [28]. Pure potassium 75 mM phosphate buffer was used as a blank. Triplicates of the diluted sample, standard or blank (25 μL), followed by 150 μL of fluorescein working solution, were inserted into a black 96-well microplate. Afterward, the microplate was incubated at 37 °C for 15 min, and subsequently, 25 μL of AAPH solution was added to each well. The fluorescence decrease (excitation at 485 nm; emission at 510 nm) was measured for 100 min at 37 °C in the microplate reader. The results were expressed in mg of TE per g of SDG extract.

### 2.4.5. Color Analysis

The coloration of extracts was evaluated using a colorimeter (Hunter Associates Laboratory, Inc., Reston, VA, USA) equipped with a D65 light source and an angle of observation of 2° for all samples. The color was characterized with a CIELAB system to calculate the coordinates $L^*$, $a^*$ and $b^*$, which were used to calculate the chroma ($C^*$) and hue angle ($H$), according to the Equations (2) and (3), respectively. The $a^*$ represents differences in red (positive) and green (negative) colors. Positive and negative values of $b^*$ represent yellow and blue colors, respectively. The $C^*$ values denote the saturation or

purity of color. The Hue angle (*H**) value denotes 0 for redness, 90 for yellowness, 180 for greenness, and 270 for blueness.

$$C* = \sqrt{a*^2 + b*^2} \qquad (2)$$

$$H* = \arctan\left(\frac{b*}{a*}\right) \qquad (3)$$

### 2.5. Statistical Evaluation

The effects of parameters studied were evaluated using a randomized full factorial design ($2 \times 3 \times 4$) for the solvent (water and ethanol), the nominal ultrasonic power (150, 350, and 450 W), and the extraction time (1, 3, 5, and 7 min), in duplicate (Table 1). Comparison of results within each column of Table 1 was carried out considering the different solvents used (water and ethanol). The effects of the parameters on global yield, genipin, geniposide, TPC, FRAP, and ORAC were evaluated by the analysis of variance (ANOVA) using Minitab 16® software (Minitab Inc., State College, PA, USA). Tukey's test's significant differences were evaluated with a 95% confidence level (*p*-value $\leq 0.05$).

**Table 1.** Effect of the process parameters on extraction results.

| Power (w) | Time (min) | $X_0$ (wt.%) | Genipin (mg/g Extract) | Geniposide (mg/g Extract) | TPC (mg GAE/g Extract) | FRAP (mg TE/g extract) | ORAC (mg TE/g Extract) |
|---|---|---|---|---|---|---|---|
| **Water Solvent** | | | | | | | |
| 150 | 1 | 14.9 ± 0.1 [e] | 101 ± 1 [g] | 52.30 ± 0.03 [h] | 28.9 ± 0.1 [a] | 5.5 ± 0.7 [a] | 87 ± 6 [cdefg] |
| | 3 | 19 ± 1 [d] | 113 ± 1 [e] | 57 ± 1 [h] | 27 ± 2 [a] | 5 ± 1 [a] | 106 ± 13 [abcdef] |
| | 5 | 19.8 ± 0.1 [cd] | 120.8 ± 0.4 [ab] | 23.1 ± 0.2 [ij] | 19.3 ± 0.4 [b] | 5.4 ± 0.2 [a] | 89 ± 7 [bcdefg] |
| | 7 | 22.1 ± 0.3 [bc] | 121.7 ± 0.5 [a] | 19.6 ± 0.3 [jk] | 21 ± 1 [b] | 5.05 ± 0.04 [a] | 121 ± 12 [ab] |
| 300 | 1 | 24.52 ± 0.1 [ab] | 120 ± 1 [ab] | 22.2 ± 0.5 [ijk] | 22 ± 1 [b] | 6 ± 1 [a] | 113 ± 16 [abcd] |
| | 3 | 25 ± 1 [a] | 116 ± 1 [cd] | 29 ± 1 [ij] | 21.4 ± 0.4 [b] | 4.4 ± 0.6 [a] | 110 ± 9 [abcde] |
| | 5 | 24.1 ± 0.2 [ab] | 116.0 ± 0.4 [d] | 20.3 ± 0.4 [ijk] | 20.9 ± 0.3 [b] | 5.06 ± 0.04 [a] | 100 ± 3 [abcdefg] |
| | 7 | 23.4 ± 0.2 [ab] | 119 ± 1 [bc] | 18.8 ± 0.4 [jk] | 20 ± 2 [b] | 6 ± 1 [a] | 129 ± 8 [a] |
| 450 | 1 | 25.5 ± 0.7 [a] | 118.8 ± 0.5 [bc] | 31.4 ± 0.6 [i] | 21 ± 1 [b] | 6 ± 1 [a] | 110 ± 11 [abcde] |
| | 3 | 25 ± 1 [a] | 114.6 ± 0.5 [de] | 7.01 ± 0.5 [l] | 18 ± 1 [b] | 5.1 ± 0.1 [a] | 112 ± 14 [abcde] |
| | 5 | 24.9 ± 0.2 [a] | 110.3 ± 0.5 [f] | 7.1 ± 0.1 [l] | 17.9 ± 0.2 [b] | 4.9 ± 0.1 [a] | 119 ± 3 [abc] |
| | 7 | 23 ± 1 [abc] | 108.58 ± 0.04 [f] | 11 ± 3 [kl] | 21 ± 1 [b] | 5.2 ± 0.5 [a] | 126 ± 7 [a] |
| Power (w) | Time (min) | $X_0$ (%) | Genipin (mg/g Extract) | Geniposide (mg/g Extract) | TPC (mg GAE/g Extract) | FRAP (mg TE/g Extract) | ORAC (mg TE/g Extract) |
| **Ethanol Solvent** | | | | | | | |
| 150 | 1 | 2.9 ± 0.7 [i] | 71.1 ± 0.3 [l] | 247 ± 5 [c] | 8.7 ± 0.4 [c] | 15 ± 5 [b] | 73 ± 7 [g] |
| | 3 | 2.1 ± 0.2 [i] | 86.3 ± 0.2 [ij] | 198 ± 5 [g] | 10 ± 1 [c] | 15 ± 3 [b] | 72 ± 2 [g] |
| | 5 | 3.1 ± 0.7 [i] | 92 ± 1 [h] | 242 ± 5 [c] | 9 ± 1 [c] | 17 ± 2 [b] | 86 ± 13 [cdefg] |
| | 7 | 3.4 ± 0.7 [i] | 103.04 ± 0.05 [g] | 312 ± 10 [a] | 8 ± 1 [c] | 12 ± 3 [b] | 70 ± 19 [g] |
| 300 | 1 | 6.9 ± 0.6 [h] | 78.5 ± 0.7 [k] | 225 ± 0.3 [ef] | 9 ± 2 [c] | 16 ± 2 [b] | 89 ± 7 [bcdefg] |
| | 3 | 8.3 ± 0.1 [gh] | 80 ± 1 [k] | 235 ± 2 [cde] | 8.4 ± 0.3 [c] | 16 ± 2 [b] | 77 ± 8 [fg] |
| | 5 | 9.1 ± 0.2 [fgh] | 83 ± 1 [j] | 243 ± 5 [c] | 8 ± 1 [c] | 16 ± 1 [b] | 82 ± 7 [defg] |
| | 7 | 9.7 ± 0.6 [fg] | 81 ± 1 [k] | 228 ± 11 [def] | 7.8 ± 0.4 [c] | 14.8 ± 0.2 [b] | 92 ± 2 [bcdefg] |
| 450 | 1 | 7.02 ± 0.05 [h] | 85 ± 1 [ij] | 260 ± 6 [b] | 10 ± 1 [c] | 15 ± 1 [b] | 89.67 ± 0.04 [bcdefg] |
| | 3 | 9 ± 1 [fg] | 84 ± 1 [ij] | 232 ± 12 [cde] | 8.6 ± 0.3 [c] | 16 ± 1 [b] | 91 ± 3 [bcdefg] |
| | 5 | 11.3 ± 0.2 [f] | 85 ± 1 [ij] | 223 ± 12 [ef] | 7.6 ± 0.2 [c] | 16 ± 1 [b] | 80 ± 5 [efg] |
| | 7 | 12 ± 1 [f] | 87 ± 1 [i] | 223 ± 5 [f] | 7.5 ± 0.3 [c] | 16.1 ± 0.1 [b] | 87.2 ± 0.2 [cdefg] |

$X_0$: global yield; TPC: total phenolic content; FRAP: ferric-reducing ability; ORAC: oxygen radical absorbance capacity. Results expressed on dry basis. Same superscript letters in the same column indicate no significant difference (*p* < 0.05).

## 3. Results and Discussion

The use of the SFE technique as a pre-treatment of the raw material allows the extraction of nonpolar compounds. By extracting these compounds with SC-$CO_2$, it is possible to obtain a matrix concentrated in polar compounds, such as phenolic compounds, iridoids, flavonoids, and polar pigments. This practice has been employed frequently in the literature to obtain a matrix more concentrated in polar compounds, making these compounds more efficient in processes that use water or ethanol as a solvent [15–17,29]. In addition to making polar compounds more accessible, the use of SFE combined with other techniques allows applying the biorefinery concept (by the full use of the raw material). In this work, the SFE removed 89% of the oil content from unripe genipap fruit, which was composed of palmitic acid (34 mg/mg extract), stearic acid (14 mg/mg extract), linoleic acid (276 mg/mg extract), and linolenic acid (48 mg/mg extract) [15].

### 3.1. Effect of the Process Parameters on Global Yield

The experimental $X_0$ obtained in all UAE conditions are presented in Table 1, and they range from 14.9 to 25.5 wt.% for aqueous extracts and from 2.1 to 12 wt.% for ethanolic extracts. It is worth highlighting that the raw material used in this work had already been subjected to another extraction process with SC-$CO_2$. An extract was obtained with a yield of 4.6% (composed mainly of linoleic acid) [15]. These results imply that after the extraction of the non-polar fraction, there are still many polar compounds to be extracted from the unripe genipap fruit.

Analysis of variance (ANOVA, $\alpha = 0.05$) showed that the interactions between time and power ($p$-value = 0.001), power and solvent ($p$-value = 0.005), and time and solvent ($p$-value = 0.008) significantly influenced the $X_0$. The plots of such interactions, showing each extraction variable's effect, are presented in Figure 2. It is important to mention that the dotted lines do not represent trends for non-numeric variables.

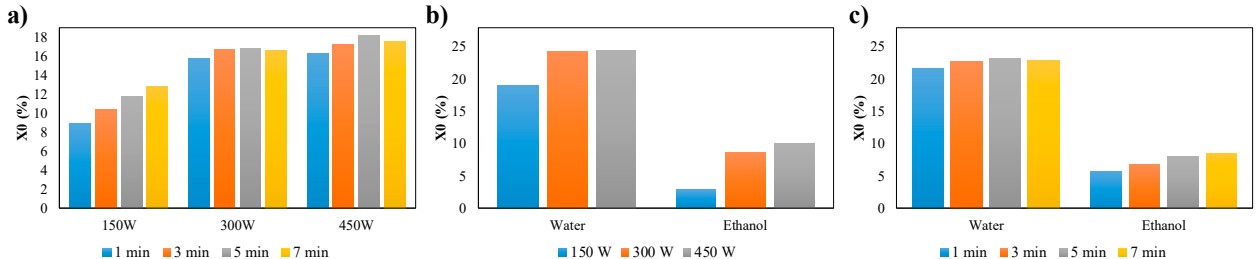

**Figure 2.** Effect of the process parameters on global yield: (**a**) time and power, (**b**) power and solvent, and (**c**) extraction time and solvent. Results expressed on a dry basis. Standard deviation by ANOVA = 0.6.

The efficiency of the extraction process depends on the nature of the compounds present in the raw material. The use of water, a solvent with higher polarity, allowed enhancing the $X_0$ compared with ethanol, as represented in Figure 2b,c. In other words, the use of water as the solvent increased $X_0$ by 4- to 8-folds at 150 W and 1- to 3-folds at 300 and 450 W (Table 1). Therefore, it can be concluded that water, especially for low ultrasound power, could access and solubilize a great number of molecules in the SDG, resulting in higher $X_0$. It is known that during the UAE, two mechanisms of mass transfer can occur: diffusion through the cell walls and washing out (rinsing) the cell contents once the walls are broken [30]. As the used material was dry and milled, once it is rehydrated, there is swelling during the steeping extraction stage, resulting in the fragmentation of the vegetal material, which increases the transfer of compounds from the raw material into the solvent [31].

The positive effect of ultrasonic power (Figure 2a,b) on the $X_0$ can be mainly attributed to the increase of energy density, which intensifies the phenomenon of acoustic cavitation [32]. This phenomenon can promote the rupture of plant cell walls, favoring the transport of the intracellular components present in the vegetable matrix into the

bulk solvent, thus increasing the $X_0$ [33]. Similarly, the positive effect of extraction time (Figure 2a,c) on the $X_0$ is attributed to the required time to desorb and solubilize the extract molecules by the solvent. For extractions using water, the equilibrium time was reached at 3 min. While for ethanol, the $X_0$ increased until 7 min (Table 1). Therefore, once again showing the extraction efficiency of water as the solvent to obtain higher $X_0$ from SDG. On the other hand, the performance over $X_0$ is not always a relevant answer parameter for evaluating the extracts, since the concentration of target compounds and biological activity provide a potential application to the extract.

### 3.2. Effect of the Process Parameters on Iridoid Content

Among the evaluated UAE variables, the interactions between time and power (*p*-value = 0.001) and time and solvent (*p*-value = 0.001) significantly influenced the genipin recovery, as shown in Figure 3a,b, respectively. The extraction of geniposide was significantly influenced only by the interaction between solvent and power (*p*-value = 0.014, Figure 3c). It is worth mentioning that the plots presented in Figure 3 show the effect of UAE variables. Experimental results for the concentration of both target compounds obtained from SDG are presented in Table 1.

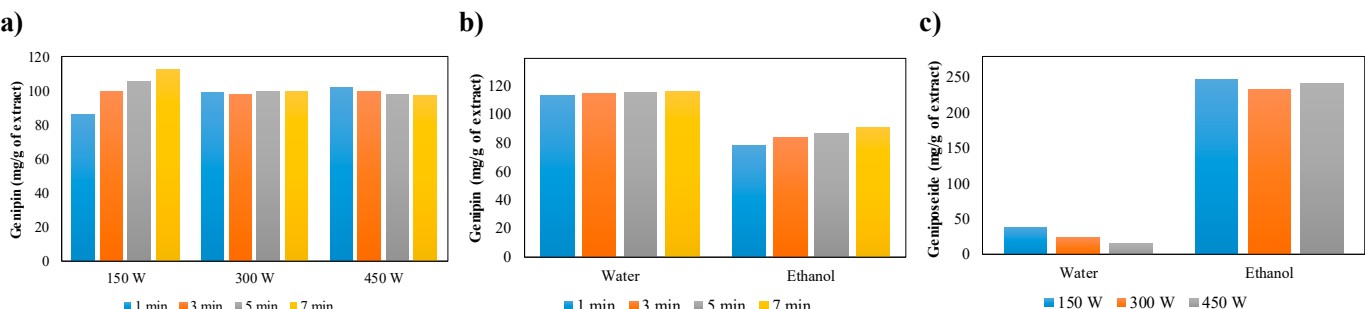

**Figure 3.** Effect of the process parameters on genipin and geniposide content: (**a**) extraction time and power on genipin content, (**b**) solvent and extraction time on genipin content, and (**c**) solvent and power on geniposide content. Results are expressed on a dry basis. Standard deviation by ANOVA for genipin = 0.6 and for geniposide = 5.5.

The highest genipin content (121.7 mg/g extract or 26.9 mg/SDG) was recovered at 150 W in 7 min using water as the solvent. This result is comparable with that extracted in our previous work from the whole unripe genipap without peel applying the low-pressure solvent extraction (LPSE) method with water at 40 °C and 1 bar for 25 min (233 mg/g extract) [34], and 81–196 mg/g unripe genipap obtained from crude genipap extracted with enzymes for 150 min [10]. It is worth mentioning that the extraction times used by these authors were much higher than the times used in this study. Considering that 7 min is a short time to extract bioactive compounds from a by-product plant matrix, the UAE was efficient in recovering such a target compound.

Regarding the ultrasonic power, different effects were observed. For instance, for water used as a solvent, as the power increased from 300 to 450 W and the extraction time increased, a reduction in the genipin content was observed. These findings are probably due to excessive cavitation and time exposed, which contributed to its degradation. Additionally, the water showed higher genipin content than ethanol, corroborating the results of $X_0$. A recent patent emphasized the use of water or other polar solvents for the obtaining of genipin-rich extracts. Indeed, non-polar organic solvents may be used as an alternative to water since their solubility is less than about 30% of the water solubility. Their polarity index varies from 0 to 5.0 [35]. However, the use of water as a solvent offers several benefits. Water is an environmentally friendly solvent, abundant, cheaper, non-toxic, and especially for extracts with food claim application, water is fully compatible.

On the other hand, the UAE with ethanol enhanced the geniposide content in ethanolic extracts (198–312 mg/g of extract), which was comparable to that from the mesocarp of crude unripe genipap (127 mg/g extract) obtained with pressurized ethanol [13]. Such

finding indicates that the fruit water content may also impair the geniposide recovery in addition to the ultrasound's positive effect. The moisture content of the raw material used in the mentioned work was 80%, which leads to the formation of a hydroalcoholic complex inside the extraction system that probably decreased the geniposide recovery. The 14 wt.% moisture content of SDG explains the high recoveries of target compounds due to the reduced competition with the water bonded within the plant matrix. The enhancement of geniposide with ethanol was attributed to the affinity in such a solvent. The sugar moiety of the geniposide structure is another factor that explains its affinity to ethanol. The authors of [22,36,37] used enzymes such as pectinesterases and β-glucosidases to hydrolyze such sugar moiety to transform the glucose from geniposide to genipin. The cost of the process increased because of the use of enzymes. Besides, studies on the application of geniposide as a natural colorant should be supported.

From the findings presented in this work, it can be concluded that performing the SC-CO$_2$ extraction before the extraction of genipin and geniposides from unripe genipap fruit by UAE does not promote the degradation of these compounds. Thus, it is important to highlight that a sequential UAE could be applied to recover at least three extract fractions. A non-polar extract was obtained by SFE, a genipin-rich extract employing water as the solvent, and a second geniposide-rich extract obtained using ethanol as the solvent. Indeed, the sequential extraction process has been applied by several authors [16,38,39] to promote the best use of raw materials through their integral use.

### 3.3. Effect of the Process Parameters on Color

The color parameters of SDG extracts are presented in Table 2. The values of chroma (*C*\*) detected in the aqueous extracts (*C*\* = 1.3–3.3) for most extraction conditions indicated low saturation of colorant in the extracts, in comparison with the ethanolic ones (*C*\* = 0.9–6.8). For the extractions with ethanol, high ultrasonic powers (300 and 450 W) during 7 min increased the values of *C*\*, while the opposite effect was found in the remaining conditions (Table 2). The values of *C*\* found for the aqueous and ethanolic extracts of SDG were higher than those of heartwood extracted with a 0.1 M sodium hydroxide solution [40].

The lightness (*L*\*) parameter of ethanolic extracts was higher than those observed in the aqueous extracts. However, it was lower than those detected in the concentrated extracts from roselle flowers [41]. The increasing of ultrasonic powers enhanced the values of *L*\*, resulting in extracts with a more transparent aspect (Table 2).

Regarding the extractions with water, most extraction conditions were associated with color in the blue region (*b*\* negative, Table 2), similarly to those detected in the endocarp + seeds of genipap extracted with pressurized ethanol [13]. The hue angle (*H*) for most aqueous extracts (300–360°—blue region) was comparable with those detected in the extracts of *Spirulina platensis* [42].

The color region of ethanolic extracts was between green (negative values for *a*\*) and yellow (positive values for *b*\*). The increase in the power increased the values of *b*\* and decreased the hue angle (Table 2).

The blue color present in the aqueous extract is related to the presence of genipin, whereas the green color of the extract obtained with ethanol is mainly due to the presence of geniposide. The blue color of the aqueous extract (Table 2) shows that the genipin present in the raw material reacted with the proteins solubilized in the extract to form the blue pigment. However, HPLC analysis revealed that the extracts had a high content of genipin (ranged from 101 to 121.7 mg/g of extract), which indicates that not all genipin reacted with the proteins present in the extract to form the blue pigment. These results are promising because they allow obtaining an extract with natural blue pigments rich in genipin that could be used to manufacture food (candies, gums, dairy beverages) and pharmaceutical products.

**Table 2.** Color parameters for the SDG extracts.

| | Water | | | | | |
|---|---|---|---|---|---|---|
| Power (W) | Time (min) | $L^*$ | $C^*$ | $H$ | $a^*$ | $b^*$ |
| 150 | 1 | 1.91 ± 0.03 | 1.4 ± 0.1 | 333.5 ± 1.3 | 1.3 ± 0.1 | −0.64 ± 0.05 |
| | 3 | 2.2 ± 0.4 | 1.5 ± 0.1 | 332.3± 1.7 | 1.4 ± 0.1 | −0.72 ± 0.05 |
| | 5 | 2.02 ± 0.03 | 1.45 ± 0.01 | 324 ± 2 | 1.18 ± 0.03 | −0.85 ± 0.03 |
| | 7 | 1.42 ± 0.14 | 1.3 ± 0.1 | 347 ± 12 | 1.25 ± 0.04 | −0.3 ± 0.03 |
| 300 | 1 | 1.9 ± 0.1 | 1.7 ± 0.1 | 319 ± 6 | 1.3 ± 0.1 | −1.1 ± 0.2 |
| | 3 | 4.3 ± 0.1 | 3.8 ± 0.1 | 106 ± 2 | 1.03 ± 0.09 | −1.6 ± 0.1 |
| | 5 | 3.9 ± 0.9 | 3.2 ± 0.1 | 110 ± 2 | 1.01 ± 0.09 | −1.5 ± 0.1 |
| | 7 | 3.0 ± 0.8 | 2.3 ± 0.3 | 312 ± 8 | 1.5 ± 0.1 | −1.3 ± 0.4 |
| 450 | 1 | 2.1 ± 0.1 | 1.6 ± 0.1 | 333 ± 4 | 1.4 ± 0.02 | −1.7 ± 0.1 |
| | 3 | 3.9 ± 0.2 | 2.7 ± 0.5 | 302 ± 2 | 1.4 ± 0.3 | −2.3 ± 0.4 |
| | 5 | 3.5 ± 0.3 | 3.3 ± 0.6 | 301 ± 6 | 1.7 ± 0.2 | −2.8 ± 0.7 |
| | 7 | 3.44 ± 0.04 | 2.6 ± 0.1 | 306 ± 1 | 1.51 ± 0.04 | −2.1 ± 0.1 |
| | Ethanol | | | | | |
| Power (W) | Time (min) | $L^*$ | $C^*$ | $H$ | $a^*$ | $b^*$ |
| 150 | 1 | 4.0 ± 0.2 | 1.2 ± 0.2 | 110 ± 4 | −0.4 ± 0.2 | 1.2 ± 0.2 |
| | 3 | 5.14 ± 0.02 | 1.0 ± 0.1 | 101 ± 3 | −0.2 ± 0.1 | 1.0± 0.1 |
| | 5 | 4.7 ± 0.3 | 1.7 ± 0.3 | 114 ± 4 | −0.7 ± 0.1 | 1.6 ± 0.3 |
| | 7 | 3.19 ± 0.01 | 0.9 ± 0.1 | 105 ± 3 | −0.2 ± 0.1 | 1.9± 0.1 |
| 300 | 1 | 4.3 ± 0.2 | 3.2 ± 0.2 | 105 ± 2 | −0.8 ± 0.1 | 3.1 ± 0.1 |
| | 3 | 4.4 ± 0.2 | 3.8 ± 0.2 | 107 ± 2 | −0.9 ± 0.2 | 3.6 ± 0.3 |
| | 5 | 5.7 ± 0.3 | 4.1 ± 0.3 | 109 ± 3 | −1.29 ± 0.07 | 3.9 ± 0.4 |
| | 7 | 5.03 ± 0.03 | 4.79 ± 0.03 | 104.1 ± 0.1 | −1.17 ± 0.02 | 4.65± 0.02 |
| 450 | 1 | 5.6 ± 0.9 | 3.8 ± 0.5 | 110 ± 1 | −1.3 ± 0.2 | 3.5 ± 0.4 |
| | 3 | 4.6 ± 1.1 | 2.6 ± 0.1 | 109 ± 5 | −0.8 ± 0.2 | 2.4 ± 0.1 |
| | 5 | 6.6 ± 1.3 | 4.6 ± 0.5 | 107 ± 5 | −1.0 ± 0.2 | 5.4 ± 0.3 |
| | 7 | 8.1 ± 0.9 | 6.8 ± 0.1 | 106.7 ± 0.4 | −1.9 ± 0.1 | 6.5 ± 0.1 |

$L^*$: luminosity: black ($L^* = 0$) and white ($L^* = 100$); $a^*$: green color (−) and red color (+); $b^*$: blue color (−) and yellow color (+); $C^*$: croma; $H^*$: hue angle.

### 3.4. Effect of the Process Parameters on the Total Phenolic Content and Antioxidant Capacity

This work aimed to evaluate the extraction process conditions to obtain high amounts of genipin and geniposide. However, in addition to these iridoids, extracts obtained from unripe genipap fruit also present phenolic compounds and antioxidant capacity, measured in this work by FRAP and ORAC.

The interactions between time and solvent ($p$-value = 0.002) and power and solvent ($p$-value = 0.001) significantly influenced the TPC, as shown in Figure 4. The antioxidant capacities measured by FRAP ($p$-value = 0.0001) and ORAC ($p$-value = 0.0001) were significantly influenced only by the solvent, as shown in Figure 5. The TPC of extractions with water (17.9–28.9 mg GAE/g extract, Table 1) were higher than those detected in extractions with ethanol (7.5–10 mg GAE/g extract.). Similar effects of solvents were detected in the X0, genipin content, and ORAC assay (87–129 mg TE/g extract for water 70–92 mg Trolox TE/g extract for ethanol, Table 1). The opposite solvent effect was found in the geniposide recovered (7–57 mg/g extract for water and 198–312 mg/g extract for ethanol) and the FRAP assay (4.4–6 mg TE/g extract for water and 14.8–17 mg TE/g extract for ethanol).

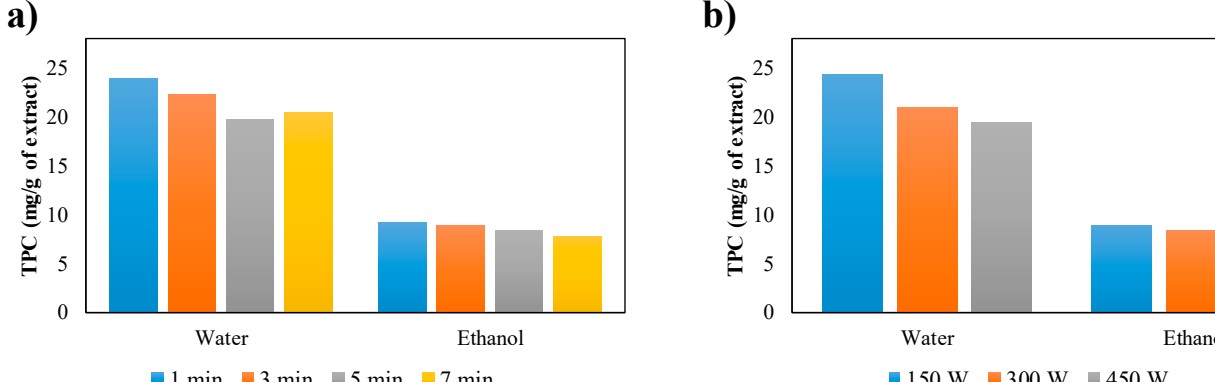

**Figure 4.** Effect of the process parameters on TPC: (**a**) solvent and extraction time and (**b**) solvent and power. Results expressed on a dry basis. Standard deviation by ANOVA for TPC = 0.9.

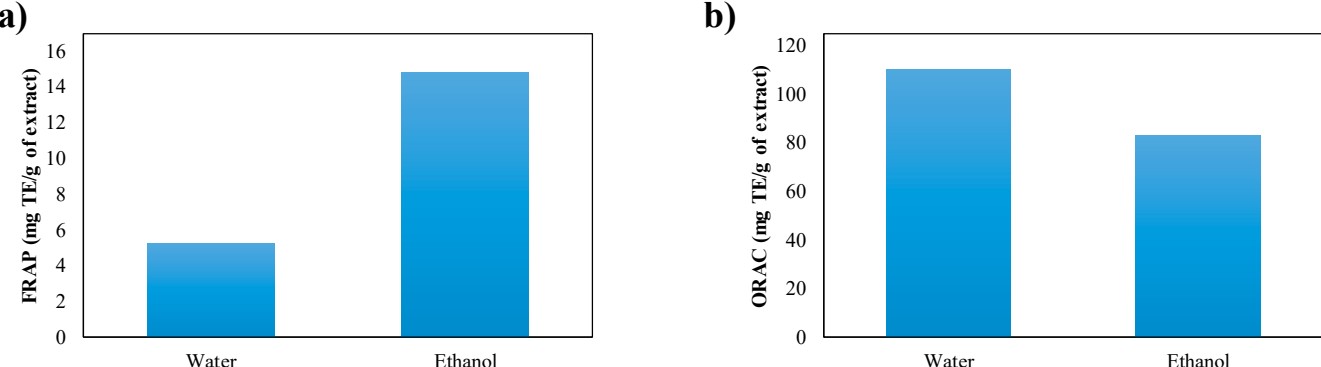

**Figure 5.** Effect of the solvent on antioxidant capacity: (**a**) effect of the solvent on FRAP and (**b**) effect of the solvent on ORAC. Results are expressed on a dry basis. Standard deviation by ANOVA for FRAP = 1 and for ORAC = 9.

The TPCs observed in the extracts were higher than those found in UAE extracts (4 mg GAE/g) and pressurized ethanol extracts (7–17 mg GAE/g of extract) [13,43], both from unripe crude genipap. For the FRAP, the antioxidant capacity of extracts was comparable with those obtained from the UAE of blue butterfly pea flower [44].

Based on the results provided in this section, it is possible to state that in addition to the color attributes, the SDG extracts have in their composition phenolic compounds, and the whole crude extract presents antioxidant capacities through ORAC and FRAP assays. These features reinforce the claim of the extract for food applications.

## 4. Conclusions

The UAE, up to 7 min, has shown as an efficient, fast, and environment-friendly method to recover natural antioxidant pigments from semi-defatted unripe genipap by-products, with results comparable to those obtained in the literature for crude unripe genipap.

Extractions with water preferentially extracted genipin, resulting in blue-colored extracts, whereas extraction with ethanol produced green-colored extracts, attributed to the enhancement in geniposide obtention. The TPC of the extracts was higher using water as the solvent, and the ultrasonic power presented a negative effect on this response.

The antioxidant capacity of the aqueous extracts was higher than that of the ethanolic extracts for the ORAC assay. In contrast, for the FRAP assay, the ethanolic extracts presented higher antioxidant capacity. The UAE was shown to be an effective technique to obtain extracts rich in iridoids. Our results are expected to valorize genipap by-products and further research the application of UAE extracts in food products and biological systems.

**Author Contributions:** Conceptualization: G.N.-N. and M.A.A.M.; Methodology: G.N.-N., Á.L.S., J.V. and J.M.; Writing—original draft preparation: G.N.-N. and Á.L.S.; Writing—review and editing: G.N.-N., Á.L.S., J.V., J.M. and M.A.A.M.; Supervision: M.A.A.M.; Project administration, M.A.A.M.; Funding Acquisition, M.A.A.M. All authors have read and agreed to the published version of the manuscript.

**Funding:** G. Náthia-Neves and Ádina L. Santana thank CAPES— This study was financed in part by the Coordenação de Aperfeiçoamento de Pessoal de Nível Superior—Brazil (Finance Code 001), for a Ph.D. and post-doctoral financial assistantships. M. Angela A. Meireles thanks CNPq for the productivity grant (309825/2020-2).

**Institutional Review Board Statement:** Not applicable.

**Informed Consent Statement:** Not applicable.

**Conflicts of Interest:** The authors confirm that this article's content has no conflict of interest.

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
