# Peer review of "Ultrasound-Assisted Extraction of Semi-Defatted Unripe Genipap (Genipa americana L.): Selective Conditions for the Recovery of Natural Colorants"

_processes, doi:10.3390/pr9081435_

Round 1
Reviewer 1 Report
Referee report for the manuscript processes-132231-peer-review-v.1:
„Ultrasound-assisted extraction of semi-defatted unripe genipap (Genipa americana L.) by super-critical CO2: selective conditions for the recovery of natural colorants”
by Grazielle Náthia-Neves, Adina L. Santana et al.
The manuscript describes the use of ultrasound-assisted extraction of semi-defatted unripe genipap to extract the two (blue and green, respectively) natural colorants genipin and geniposide.
Other than the title lets expect, the extraction is not performed with supercritical CO2, but with water or ethanol. It is suggested to rephrase the title to avoid misunderstandings. The use of ultrasonic extraction is not entirely new (as claimed in l.89 by the authors), as there has been at least one precedent of this technique applied to the extraction of genipin from genipap which the authors have quoted as reference [22]. Also, the use of SFE for defatting of food products prior to further extraction or sample preparation steps is a common procedure.
The authors have reported on their experimental work which is mainly the combination of these two sample preparation steps and the subsequent optimization of the extraction conditions with respect to the extraction sol vent, extraction duration and ultrasound power used. This is relevant from the point of view of applied / processing technology, but provides only incremental novel findings from a scientific point of view.
The manuscript is well and clearly written. It is sound from the technical point of view; experiments have been appropriately designed and evaluated. There is no important criticism to the content of the paper, except for some formal aspects:
- Equations (1) – (3) will have to be reformatted – I guess this is the work of the publisher.
- 164: “afterward” -> “afterwards”
- Table headings (and figure captions) are in larger fonts than the main text. I assume that also this will be corrected by the publisher.
- In Table 1, the figures and also the main text, X0 is used frequently without explaining what this symbol stands for.
- Table 1: The meaning of the superscript letters does not become clear, or at least appears not to be consistent with what is claimed.
- Table 1 (also true for table 2): Results should be reported with the same number of significant digits, e.g.., not as 6±1, and for another result as 5.06±0.04.
- Figures of same style should also be reproduced in the same size (cf. figures 2 and 3 !)
- The figure type for figures 2 b and c and also for Figures 3 b and c is considered unsuitable: A line graph always suggests a (numeric) trend and should thus be used only if the ordinate is a number scale. If the ordinate is a category scale (water / ethanol), a line graph is unsuitable and should be replaced by, for example, a bar graph.
- Figure caption of Figure 2: It Is claimed in the figure caption that the interaction between the two parameters is depicted in each of the graphs Figure 2b and 2 c, respectively. I disagree here, as I would define an interaction (in the mathematical sense) the dependence of the result on both parameters X1 and X2: Y = a * X1 * X2. What is depicted in the graphs 2a and 2b, however, is only the change of Extraction yield for the two solvents investigated at different US power or extraction time.
- 325: is the result extracted in the Authors’ precious work ? Rephrase !
- 328: Please reconsider the numeric accuracy of the results reported here: 80.59 – 195.97 ! (two decimal places appears exaggerated considering that the low and the high end are more than a factor 2 apart !)
- 331: The extraction times have probably been used by the study authors, but not by the articles ! Rephrase !
- Figures 4 and 5: Same comment as made before for figures 1 and 2. The use of bar graphs would also allow to report error bars which are missing in this type of presentation !
- 423-429: Paragraph was moved so far to the right that it was partially illegible !
Reviewer 2 Report
The manuscript titled „Ultrasound-assisted extraction of semi-defatted unripe genipap (Genipa americana L.) by supercritical CO2: selective conditions for the recovery of natural colorants“ is related to the recovery of iridoids and polyphenols from genipap. The authors proposed sequel extraction of non-polar and polar bioactives from unripe genipap using SFE and UAE (with water and ethanol). The manuscript is generally well written, significant extraction parameters are well explained and extraction efficiency is substantially high. There some minor issues which could be addressed:
- Please explain why unripe fruit was used? Is that fruit removed from the trees to enhance sugar accumulation in remaining fruits, thus it is considered as agricultural by-product. Or maybe because concentration of colourants is higher in unripe fruits?
- Please explain why didn’t you use maceration as comparative method to UAE?
Typos:
Page 7 Lines 254-256 – I suppose that correct units are mg/g extract
Page 10 Line 314 - (p-value = 0.014, Figure 3c)
